# Alterations in Protein Translation and Carboxylic Acid Catabolic Processes in Diabetic Kidney Disease

**DOI:** 10.3390/cells11071166

**Published:** 2022-03-30

**Authors:** Kimberly S. Collins, Michael T. Eadon, Ying-Hua Cheng, Daria Barwinska, Ricardo Melo Ferreira, Thomas W. McCarthy, Danielle Janosevic, Farooq Syed, Bernhard Maier, Tarek M. El-Achkar, Katherine J. Kelly, Carrie L. Phillips, Takashi Hato, Timothy A. Sutton, Pierre C. Dagher

**Affiliations:** 1Division of Nephrology and Clinical Pharmacology, Department of Medicine, Indiana University School of Medicine, Indianapolis, IN 46202, USA; kscollins@c-path.org (K.S.C.); meadon@iupui.edu (M.T.E.); rimelof@iu.edu (R.M.F.); 2Division of Nephrology, Department of Medicine, Indiana University School of Medicine, Indianapolis, IN 46202, USA; yicheng@iu.edu (Y.-H.C.); dbarwins@iu.edu (D.B.); twmccart@iu.edu (T.W.M.); djanosev@iupui.edu (D.J.); bfmaier@iu.edu (B.M.); telachka@iu.edu (T.M.E.-A.); kajkelly@iu.edu (K.J.K.); thato@iu.edu (T.H.); 3Department of Pediatrics and Herman B. Wells Center, Indiana University School of Medicine, Indianapolis, IN 46202, USA; fsyed@iupui.edu; 4Department of Pathology, Indiana University School of Medicine, Indianapolis, IN 46202, USA; cphilli3@iupui.edu

**Keywords:** gene expression signature, single nuclear RNA sequencing, metabolomics, ribosomal profiling

## Abstract

Diabetic kidney disease (DKD) remains the leading cause of end-stage kidney disease despite decades of study. Alterations in the glomerulus and kidney tubules both contribute to the pathogenesis of DKD although the majority of investigative efforts have focused on the glomerulus. We sought to examine the differential expression signature of human DKD in the glomerulus and proximal tubule and corroborate our findings in the db/db mouse model of diabetes. A transcriptogram network analysis of RNAseq data from laser microdissected (LMD) human glomerulus and proximal tubule of DKD and reference nephrectomy samples revealed enriched pathways including rhodopsin-like receptors, olfactory signaling, and ribosome (protein translation) in the proximal tubule of human DKD biopsy samples. The translation pathway was also enriched in the glomerulus. Increased translation in diabetic kidneys was validated using polyribosomal profiling in the db/db mouse model of diabetes. Using single nuclear RNA sequencing (snRNAseq) of kidneys from db/db mice, we prioritized additional pathways identified in human DKD. The top overlapping pathway identified in the murine snRNAseq proximal tubule clusters and the human LMD proximal tubule compartment was carboxylic acid catabolism. Using ultra-performance liquid chromatography–mass spectrometry, the fatty acid catabolism pathway was also found to be dysregulated in the db/db mouse model. The Acetyl-CoA metabolite was down-regulated in db/db mice, aligning with the human differential expression of the genes ACOX1 and ACACB. In summary, our findings demonstrate that proximal tubular alterations in protein translation and carboxylic acid catabolism are key features in both human and murine DKD.

## 1. Introduction

Diabetes mellitus (DM) afflicts over 400 million people worldwide [1] and over 30 million in the US [2]. Diabetic kidney disease (DKD) is one of the most serious complications of DM, affecting 19 million people in the United States [3], and is the leading cause of end-stage kidney disease (ESKD) [4]. DKD is classically viewed as a glomerular disease; however, focusing solely on glomerular pathology as the primary site of injury overlooks key biological alterations in the tubulointerstitial compartment that contribute to the pathophysiology of DKD [5]. Indeed, the prognosis and progression of DKD frequently parallel tubular atrophy and the extent of interstitial disease [6,7]. Further germane to this point is evidence that targeting the proximal tubule with SGLT2 inhibitors slows the progression of DKD [8].

The cellular mechanisms leading to the development and progression of DKD are admittedly complex. The diabetic milieu is characterized by metabolic abnormalities, inflammation, oxidative stress, and alterations in protein homeostasis. All of these derangements contribute to cell stress and the development of DKD. There is growing evidence that altered translation, the fundamental step in gene expression that creates functional proteins, is important in both responding and contributing to cell stress [9,10]. Alterations in translation contribute to imbalances in protein homeostasis and impact human health and disease [11,12,13,14].

Interrogation of human kidney biopsy specimens routinely utilizes light microscopy, transmission electron microscopy, and immunoreactions to enable the pathologist to render a diagnosis and prognosis. However, these diagnostic tools have limitations in uncovering the molecular mechanisms underlying kidney disease including DKD. Consequently, animal models of DM have historically been used to study the pathophysiological mechanisms of DKD; however, none of these models appropriately simulate human DKD [15]. Although recent efforts to apply more advanced molecular interrogation techniques to human kidney biopsy specimens are encouraging [16], we believe using animal models in parallel with human kidney tissue remains an important experimental approach to provide unique insights into the molecular underpinnings of DKD.

We sought to examine the differential expression signature of human DKD in the glomerulus and proximal tubule. We hypothesized that the proximal tubular transcriptomic signature, in addition to that of the glomerulus, also determines DKD outcomes. We addressed this hypothesis by interrogating human kidney biopsy specimens with laser microdissection of glomerular and proximal tubular regions to identify pathways associated with DKD and its progression. Next, key pathways were corroborated in a diabetic mouse model using a multi-modal approach of single nuclear RNA sequencing (snRNAseq), polyribosomal profiling, and metabolomics. In addition to derangements in protein translation, our findings support a potential role of carboxylic acid metabolism as a key factor in diabetic kidney disease progression.

## 2. Materials and Methods

### 2.1. Human Subjects

This study was approved by the Institutional Review Board of Indiana University School of Medicine (IRB no. 190657223). Twenty-seven kidney samples were obtained from the Kidney Precision Medicine Project Consortium and the Biopsy Biobank Cohort of Indiana [17]. Kidney biopsies were indication biopsies for proteinuria or a decline in renal function in subjects with diabetes mellitus. Eighteen samples were acquired from adults with diabetic kidney disease and without a second glomerular lesion. Nine reference samples without histologic evidence of DKD were acquired from non-neoplastic parenchyma of nephrectomy specimens or deceased donors. Clinical and histopathologic variables were extracted from the electronic health record. The baseline estimated glomerular filtration rate (eGFR) was defined as the value closest to the date of biopsy unless a 20% or greater decline in eGFR precipitated biopsy referral. In this scenario, an eGFR within 1 year prior to biopsy and before the 20% decline was set as the baseline for progression calculations.

### 2.2. Laser Microdissection of Human Kidney Tissue and RNA-Sequencing

Cryosections of 12 μm thickness were cut from frozen tissue blocks preserved in Optimal Cutting Temperature medium, adhered to polyphenylene sulfide (PPS) membrane slides, and processed using a Rapid Stain protocol as previously described [18,19]. A minimum area of 500,000 μm^2^ was dissected for glomerular and proximal tubular compartments using the pulsed UV laser on the Leica LMD6500 and a 20× objective. Dissected tissue was collected in a sterile RNAse-free tube containing RNA Extraction Buffer and RNA was isolated according to the manufacturer’s instructions (Arcturus PicoPure RNA Isolation Kit, ThermoFisher, Waltham, MA, USA). RNA quality was determined using Agilent 2100 Bioanalyzer.

Sequencing was performed at the Indiana University Center for Medical Genomics Core. Ribosomal RNA was depleted using the RiboGone—Mammalian Kit protocol (Cat #634847, Takara Bio USA, Mountain View, CA, USA). The SMARTer Universal Low Input RNA Kit protocol v2 (Cat #634938, Takara Bio USA, Mountain View, CA, USA) was used for cDNA synthesis and library construction. Sequencing was performed with 2 × 75 bp paired-end configuration on the Illumina HiSeq 4000 using the HiSeq 3000/4000 PE SBS Kit, and the sequenced data were mapped to the hg38 genome using STAR. Uniquely mapped sequencing reads were assigned to the hg38 reference genome genes using Rsubread featureCounts [20,21].

### 2.3. Animal Study Approvals

All animal protocols were approved by the Indiana University Institutional Animal Care Committee and conform to the National Institutes of Health Guide for the Care and Use of Laboratory Animals. For bulk RNA-sequencing, polyribosomal profiling, and metabolomics, male mice strains C57BL/6J (#000664, background control/C57 mice) and B6.BKS–Leprdb/J (#000697, db mice) were obtained from the Jackson Laboratory (Bar Harbor, ME, USA). Mice were aged 9–12 weeks and weighed ~30 g (C57 mice) or 40–50 g (db mice). Mice were sacrificed and kidneys harvested (*n* = 4 or 5 per group).

### 2.4. Isolation of Mouse Kidney Tissue and Bulk RNA-Sequencing

Kidneys were snap-frozen and RNA was extracted using a QIAGEN RNeasy Plus Midi Kit with a genomic DNA removal column. RNA quality was determined using an Agilent 2100 Bioanalyzer. The Illumina TruSeq Stranded mRNA Library Prep Kit was used for library construction. Sequencing was performed with 2 × 75 bp paired-end configuration on Illumina HiSeq 4000 using the HiSeq 3000/4000 PE SBS Kit, and the sequenced data were mapped to the mm10 genome using STAR. Uniquely mapped sequencing reads were assigned to the mm10 reference genome genes using Rsubread featureCounts.

### 2.5. Isolation of Mouse Kidney Tissue and Single-Nuclear RNA-Sequencing

Mice aged 5, 8, 11, 16, and 20 weeks were evaluated (*n* = 5 total, BKS.Cg-Dock7^m^+/+Lepr^db^/J #000642). Heterozygotes from the same strain were used as background controls (*n* = 3). Urine albumin was measured by a mouse albumin ELISA kit (Bethyl Laboratories, E99-134, Montgomery, TX, USA) and urine creatinine was measured by a QuantiChrom creatinine assay kit (Bioassay System, DICT-500, Hayward, CA, USA). On ice, a portion of each kidney was added to 1 mL of Nuclei EZ lysis buffer supplemented with 2% protease inhibitor (Thermofisher), and 1% Superase (Thermofisher). After mincing, 1 mL of buffer was added and tissue was homogenized using the KONTES Dounce Tissue Grinder (Kimble Chase, Rockwood, TN, USA). Homogenate was gently mixed with 2 mL of buffer and incubated on ice for 5 min. Homogenate was filtered through a 40 μm strainer and treated with lysis buffer supplemented with 1% Superase, then filtered again through a 30 μm strainer, and subsequently resuspended in PBS supplemented with 2% BSA and 1% Superase. This suspension was then filtered through a 5 μm filter and the concentration was adjusted to 1–3 million nuclei/mL prior to submission for sequencing.

Single nuclear 3′ RNA-sequencing was performed at the Indiana University Center for Medical Genomics Core using the Chromium single-cell system version 3 (10× Genomics, San Francisco, CA, USA) and the NovaSeq6000 sequencer (Illumina, San Diego, CA, USA). Cell Ranger 4.0 was utilized to generate sample-specific FASTQ files and reads were aligned to the mm10 reference genome using STAR. Seurat v 3.0.1 was used to integrate samples using the following quality control metrics: included gene counts between 200–3000 and percent mitochondrial gene less than 50%. In total, 58,405 cells were retained for downstream analysis. Standard preprocessing, feature selection, dimension reduction (20 principal components), identification of anchors between samples, and integration were performed according to Seurat v3 anchoring methods. Clusters were annotated based on common gene expression markers [22] and using Kidney Cell Explorer [23].

### 2.6. Isolation of Mouse Kidney Tissues and Polyribosomal Profiling

Background control and db mice were sacrificed at 9 and 12 weeks of age. Cardiac perfusion via the left ventricle was performed with 6 mL of cycloheximide (100 μg/mL in PBS). Harvested kidneys were immediately placed in a lysis buffer consisting of 1% Triton X-100, 0.1% deoxycholate, 20 mM Tris-HCl, 100 mM NaCl, 10 mM MgCl_2_, EDTA-free Protease Inhibitor Cocktail Tablet (Roche, Penzberg, Germany), 50 μg/mL cycloheximide, and RNAsin (1:500 dilution). Tissues were homogenized at 4 °C using a Precellys tissue homogenizer (Precellys, Montigny-le-Bretonneux, France). Tissue homogenates were incubated on ice for 10 min, then centrifuged at 9600× *g* for 10 min. The supernatant was added to the top of a sucrose gradient generated by BioComp Gradient Master (10% sucrose on top of 50% sucrose in 20 mM Tris-HCl, 100 mM NaCl, 5 mM MgCl_2_, and 50 mg/mL cycloheximide) and centrifuged at 284,000× *g* for 2 h at 4 °C. The gradients were harvested from the top in a Biocomp harvester (Biocomp Instruments, Fredericton, NB, Canada), and the RNA content of eluted ribosomal fractions was continuously monitored with UV absorbance at 254 nm.

### 2.7. Metabolomics

To assess metabolomic signatures, samples of background control and db mice (*n* = 5 per group) were prepared using the automated MicroLab STAR^®^ system (Hamilton Company, Reno, NV, USA). Metabolomic analysis was performed at Metabolon Inc. Briefly, snap-frozen kidney tissues were processed following the Metabolon standard extraction method as per company protocol. The extracts were analyzed on a Waters ACQUITY ultra-performance liquid chromatography (UPLC) with a C18 column (Waters UPLC BEH C18-2.1 × 100 mm, 1.7 µm) and a Thermo Scientific Q-Exactive high resolution/accurate mass spectrometer interfaced with a heated electrospray ionization (HESI-II) source and Orbitrap mass analyzer operated at 35,000 mass resolution.

Raw data were extracted, peak-identified and QC processed using Metabolon’s (Durham, NC, USA) hardware and software. Peaks were quantified using area-under-the-curve. Standard statistical analyses with Welch’s two-sample t-test was performed in ArrayStudio on log-transformed data.

### 2.8. Differential Gene Expression, Pathway Analysis, and Statistics

Human LMD and mouse kidney tissue expression data from RNA-sequencing was quantile normalized and differential expression was determined using an exact test in edgeR with *p* values < 0.05 considered statistically significant after Benjamini–Hochberg false discovery rate (FDR) multiple testing correction. For mouse kidney samples that underwent snRNA-sequencing, differential expression was determined using a Wilcoxon Rank Sum test with Bonferroni adjusted *p* values < 0.05 considered statistically significant. However, uncorrected *p*-values are provided in the manuscript.

Gene expression data were investigated for enriched pathways in Gene Ontology, Kegg, and Reactome using a transcriptogram network analysis as previously described [19,24,25,26]. Briefly, all protein-coding genes, regardless of p-value, are ordered by protein–protein interaction networks. A Monte Carlo algorithm was used to cluster genes by shared biological functions. Significance is determined using a two-tailed Welch’s *t*-test after 500 random permutations to estimate the false discovery rate with *p* values < 0.05 considered statistically significant.

Since only a subset of genes is expressed in any given cell for snRNA-seq data, the transcriptogram method is not feasible for this data type. Pathway analysis for snRNA-seq involved enrichment of differentially expressed genes in Gene Ontology, Kegg, and Reactome according to Fisher’s exact test, without accounting for protein–protein network interactions. All proximal tubular cell clusters across all time points were merged to identify enriched genes between control and db mice. The single podocyte cluster was also merged across time points. This method was applied to human RNA-seq data as well when assessing overlap with the snRNAseq dataset. FDR-corrected p values for pathways were used to identify overlap between matching sub-segments. PAM (partitioning around medoids) was used to cluster human diabetic samples according to the genes involved in each overlapping pathway. Differential clinical features were determined using the Wilcoxon Rank Sum test between clustered groups.

## 3. Results

### 3.1. Human Subjects

Kidney tissue samples from 18 subjects with DKD and 9 reference subjects without DKD were acquired and studied (Table 1). The mean age at tissue acquisition was 55.0 years of age in subjects with DKD and 50.0 in the reference nephrectomy subjects. Subjects with DKD were 61.1% female and 27.8% were Black. Most DKD subjects had nodular glomerulosclerosis and about half had greater than 80% effacement of podocyte foot processes. Nephrotic range proteinuria was observed in 55.6% of DKD subjects. Baseline eGFR was 63.2 ± 26.2 mL/min. Subjects lost an average of 14.5 mL/min/year of eGFR over follow-up duration, which was 47.7 months on average. As a molecular comparator group, nine reference nephrectomy samples were obtained. These samples were evaluated by a blinded renal pathologist (C.P.) and did not show histologic evidence specific for diabetic kidney disease; however, clinical and demographic data such as race, proteinuria, and eGFR were not available.

### 3.2. Differential Gene Expression and Pathway Analysis in the Diabetic Human Glomerulus and Proximal Tubule

Differential gene expression and pathway analyses were performed in the DKD and reference groups. Differentially expressed genes (DEGs) were identified in laser microdissected glomeruli and proximal tubules of both groups (Supplemental Table S1; https://doi.org/10.6084/m9.figshare.14450190.v1, accessed 23 January 2022). Known glomerular markers were down-regulated in DKD, including nephrin (NPHS1, 3-fold decrease, *p* = 1.0 × 10^−4^) and phospholipase C epsilon 1 (PLCE1, 2.56-fold decrease, *p* = 3.3 × 10^−4^). Likewise, proximal tubule markers including the sodium–hydrogen exchange cofactor 3 (PDZK1, 4.2-fold decrease, *p* = 1.9 × 10^−5^) and the sodium–phosphate cotransporter (SLC34A1, 3.2-fold decrease, *p* = 8.1 × 10^−4^) were down-regulated in DKD, likely reflecting chronic injury.

We examined pathway enrichment via a transcriptogram network analysis in the human glomerulus and proximal tubule of DKD and reference samples. This analysis incorporates the magnitude of differential expression, direction of effect, level of significance, and proximity of protein-coding genes ordered along the x-axis according to known protein–protein interactions. The most enriched pathways of the glomerulus included GTPase mediated signal transduction (*p* = 4.3 × 10^−6^) and G-protein coupled receptor (GPCR) ligand binding (*p* = 8.1 × 10^−6^). In the proximal tubule, the top pathways were Rhodopsin-like receptors (*p* = 4.1 × 10^−6^) and olfactory signaling (*p* = 1.9 × 10^−4^). Interestingly, the ribosome (translation) pathway was enriched in both the glomerulus and proximal tubule (Figure 1). A complete list of differentially regulated pathways is included in Supplemental Table S2; (https://doi.org/10.6084/m9.figshare.14450196.v1, accessed 23 January 2022)

To determine the direction of effect for translation pathways enriched in the DKD samples, the pathview schematic of the KEGG equivalent pathway, ribosome hsa03010, was compiled for both the glomerulus and proximal tubule. Genes within the translation and metabolism of amino acids and derivatives GO pathways aligned with the ribosome KEGG pathway with an overlap of 130 and 87 out of 186 genes, respectively. The overall expression of ribosomal genes was up-regulated in both the glomerulus and proximal tubules of DKD subjects (Figure 2).

### 3.3. Translation Is Altered in the Diabetic Mouse Kidney

To better understand the significance of pathway enrichment in the human DKD biopsy specimens, we queried differential expression in the db mouse as compared to a background control strain using bulk RNA sequencing. Pathway analysis (at 9 weeks of age) between the diabetic mice versus the control strain revealed the top enriched pathway was cGMP-protein kinase G signaling (*p* = 2.0 × 10^−6^) (Figure 3A). Among the enriched pathways was translation-related ribosome biogenesis (*p* = 0.047), which “overlapped” with the human DKD pathway enrichment dataset.

To directly examine the functional consequence of increased transcription and expression of proteins regulating translation, we performed polyribosomal profiling on kidneys from db and background control mice. This assay quantifies the general level of translation based on mRNA occupancy by ribosomes. Polyribosomal profiling demonstrated that the polysome-to-monosome area-under-the-curve ratio is increased in diabetic mice (9.76) compared to control mice (7.73, *p* < 0.05 for the difference of ratios), consistent with increased global translation in the kidney in diabetes (Figure 3B).

We attempted to correlate the glomerular and proximal tubule-specific human expression signatures to cell-type-specific signatures in the mouse model by examining snRNAseq derived expression in podocytes and proximal tubule cells. Furthermore, we examined whether these changes had a temporal association before and after the onset of albuminuria. For this purpose, we selected a db strain on a BKS background (Kallis strain) that is known to more reliably develop albuminuria as evidence of diabetic nephropathy and compared these to age-matched heterozygote controls (Figure 4). In these mice, the albumin-creatinine ratio at 16 weeks was 1.24 ± 0.60 µg Alb/µg creatinine which corresponds to previously reported values for this mouse strain at this age.

Unbiased clustering of the snRNAseq dataset yielded seventeen clusters corresponding to the expected cell types of the kidney (Figure 4A). Clusters were defined by known expression markers (Figure 4B). Differential expression between diabetic and control mice was assessed in both podocytes and in a merged set of all three proximal tubule clusters. Although a number of important translation pathway genes were differentially expressed, the overall pathways did not reach statistical significance for enrichment (Figure 4C, Supplemental Table S1; https://doi.org/10.6084/m9.figshare.14450190.v1, accessed 23 January 2022). Access to an interactive link of the snRNA seq data is provided here: https://connect.rstudio.iu.edu/content/21/, accessed 23 January 2022.

### 3.4. Single Nuclear RNA Sequencing Reveals Highly Enriched Pathways in the db Mouse Model

In the translation pathway analysis above, we first filtered potential pathways using the human datasets, then validated the relevance of translation with the mouse bulk expression signature and polyribosomal profiling. In order to broaden our search for clinically relevant pathways, we next reversed the analysis order, starting with the murine snRNAseq dataset, and then assessed the overlap within the human dataset.

DEGs of each snRNAseq cluster were identified (Supplemental Table S1; https://doi.org/10.6084/m9.figshare.14450190.v1, accessed 23 January 2022). Based on this differential expression, pathway analysis was undertaken. As mentioned above in the method section, the nature of snRNAseq data precludes a transcriptogram network analysis because individual nuclei lack sufficient diversity of expressed genes to construct a protein–protein interaction network. Thus, pathways were assessed by standard Fisher’s exact test enrichment. In the mouse glomerulus, the top altered pathways included the coenzyme metabolic process (*p*-value: 2.1 × 10^−8^), organic anion transport (*p*-value: 1.35 × 10^−7^), and small molecule catabolic process (*p*-value: 1.58 × 10^−6^; and Supplemental Table S2; https://doi.org/10.6084/m9.figshare.14450196.v1). In the mouse proximal tubular cell clusters, the top altered pathways included the coenzyme metabolic process (*p*-value: 2.59 × 10^−13^), organic anion transport (*p*-value: 6.56 × 10^−12^), and carboxylic acid catabolic process (*p*-value: 1.28 × 10^−11^; Figure 5A and Supplemental Table S2; https://doi.org/10.6084/m9.figshare.14450196.v1, accessed 23 January 2022).

We applied the same methodology to human sample enrichment between DKD and reference samples. Enriched human glomerular pathways included alpha-defensins (*p*-value: 2.26 × 10^−17^), membrane disruption (*p*-value: 1.1 × 10^−16^) and innate immune response in mucosa (*p*-value: 3.94 × 10^−15^). Enriched proximal tubule pathways were membrane disruption (*p*-value: 1.56 × 10^−16^), alpha-defensins (*p*-value: 6.33 × 10^−16^) and positive regulation of peptidyl-serine phosphorylation of STAT protein (*p*-value: 5.14 × 10^−15^; Figure 5B and Supplemental Table S2; https://doi.org/10.6084/m9.figshare.14450196.v1, accessed 23 January 2022).

Pathway overlap between the human and mouse datasets was assessed for the glomerulus and proximal tubule cell types. When comparing pathways that overlap for both human and mouse kidneys, the kidney development pathway was altered in the glomerulus and four pathways including carboxylic acid catabolic process, steroid hormone biosynthesis, glutathione metabolism, and biological oxidations were enriched within the proximal tubule cells. The top pathway identified in the murine snRNAseq that was also enriched in the human dataset was the carboxylic acid catabolic process, identified in proximal tubule cells of murine snRNAseq and the proximal tubule compartment of the human LMD transcriptomics.

### 3.5. Altered Regulation of the Carboxylic Acid Catabolic Processes in the Proximal Tubule Is Associated with Progression of Kidney Failure

As an exploratory analysis, we queried whether the clinical features of the DKD subjects in Table 1 might be associated with carboxylic catabolic acid process genes or translation pathway genes. After clustering human diabetic samples by expression of genes involved in the carboxylic acid catabolism pathway, two DKD sub-groups were identified in the principal component analysis based on gene expression (Figure 5C). Each group was then assessed for a binary outcome of progression rate, wherein “moderate progressors” were defined by a slope of ≤10 mL/min/year eGFR loss (average loss of 3.3 ± 3.8 mL/min/year) and “rapid progressors” had a slope of >10 mL/min/year eGFR loss (average loss of 23.4 ± 22.7 mL/min/year). Carboxylic acid catabolic process expression in the proximal tubule was associated with rapid eGFR loss (*p* = 0.034) (Figure 5D). The top genes driving the clustering in this pathway included ACOX1, ACADVL, ACACB, CD44, and SHMT2. Overall, SHMT2 and CD44 expression was upregulated in Group 2 whereas ACADVL, ACACB, and ACOX1 had higher expression in Group 1. Other clinical and pathological features including proteinuria, age, race, glomerular obsolescence, interstitial fibrosis, and tubular atrophy (IFTA) were not associated with carboxylic acid process catabolism. Finally, no clinical or pathological features were associated with translation pathway expression after clustering.

### 3.6. Metabolomics

We sought to further assess critical metabolites in the carboxylic acid catabolic process pathways by performing a metabolomic analysis in db and control mice. Many pathways related to carboxylic acid catabolism were dysregulated in the metabolomic dataset for fatty acid metabolism (Figure 6) and amino acid metabolism (Supplemental Figure S1; https://doi.org/10.6084/m9.figshare.14450277.v1, accessed 23 January 2022). For example, a number of acyl carnitine fatty acids and acyl choline fatty acids were up-regulated in the db mouse. In contrast, several long-chain fatty acids, such as eicosenoate, were down-regulated in the db mice. The observed dysregulation of fatty acids is potentially consistent with the human gene expression signatures, demonstrating alterations in very-long-chain specific acyl-CoA dehydrogenase (ACADVL) expression. ACADVL is an Acyl-CoA dehydrogenase and an important first step in mitochondrial fatty acid beta-oxidation.

A single metabolite was up-regulated in the ketone pathway, 3-betahydroxybutyrate. Finally, Acetyl-CoA was down-regulated in db mice, aligning with the human differential expression in ACOX1 and ACACB. Both of these metabolites are known to be differentially expressed in diabetes mellitus [27,28]. Among the amino acid metabolism pathways, most were down-regulated in db mice, except for leucine, isoleucine, and valine metabolism.

## 4. Discussion

In the present study, we employed a multi-modal omics network to explore the signature of DKD, translating observed findings across organisms and spanning bench to bedside. This approach allowed us to distill salient pathophysiologic features of DKD in humans, despite heterogeneity and a small sample size. Key dysregulation was identified in both mice and humans in the translation and carboxylic acid catabolism pathways.

Our results confirm and augment the current understanding of the molecular pathogenesis of DKD. Translation is the fundamental biologic process by which mRNA is read and converted into functional proteins. The examination of translation in the kidney during diabetes has historically focused on mTOR/AMP kinase signaling [29,30,31]. Thus, a comprehensive understanding of translation in DKD currently escapes the field. Interestingly, prior evidence underscores the importance of dysregulated translation in the murine glomerulus of DKD [32]. Our results in the db mouse align with these prior investigations, demonstrating alterations in translation pathway gene expression, corroborated by polyribosomal profiling. However, our investigation expands upon these animal studies to also identify translational dysregulation in human DKD, both in the glomerulus and importantly, in the proximal tubule. Dysregulated translation can alter the balance between protein synthesis and protein degradation. This imbalance can promote endoplasmic reticulum (ER) stress and the subsequent development and progression of DKD [33].

Analogously, our multi-omics approach identified carboxylic acid catabolism as a priority pathway in the proximal tubule in the snRNAseq dataset of the db mice and the human proximal tubule LMD dataset. Metabolomic analysis in the db mice revealed a variety of fatty acid metabolites differentially expressed in diabetes. Both 3-betahydroxybutyrate and acetyl-CoA were differentially expressed metabolites and the results observed in our study align with the expected direction of effect in diabetes mellitus [27,28].

In our exploratory analysis, five genes in the carboxylic acid catabolism pathway were found to drive clustering in human subjects according to their DKD progression. This pathway has been previously identified as dysregulated in the tubulointerstitium of human diabetic kidneys [34]. In our study, increased expression of ACADVL, ACACB, and ACOX1 were associated with reduced progression in our carboxylic acid group 1. ACADVL is an important mediator of mitochondrial fatty acid beta-oxidation. Protein levels of this acyl-CoA dehydrogenase have been shown to be decreased in the brain of diabetic mice [35]. After chronic fitness training, ACADVL expression in the muscle increases, further suggesting improved outcomes in diabetic individuals [36]. A second gene co-expressed with ACADVL was ACOX1. ACOX1 expression was shown to be up-regulated after phellinus linteus treatment, reducing blood glucose levels and improving insulin resistance in diabetic mice [37]. Finally, ACACB polymorphisms have been associated with diabetes mellitus, suggesting this gene is important in the regulation of metabolic disorders [38]. While the direction of effect for these prioritized genes is supported in the literature (i.e., up-regulation is associated with better outcomes), we are cautious to draw conclusions because both of our carboxylic acid groups had DKD.

Two genes, CD44 and SHMT2, were up-regulated in the carboxylic acid group 2, comprised of individuals with rapid progression of DKD. CD44 acts as both a receptor for hyaluronan and osteopontin, of which both molecules are associated with the pathogenesis of DM. For example, hyaluronan promotes muscle insulin resistance [39]. There is also a critical role for osteopontin in DKD. Studies have revealed that osteopontin expression in DKD mouse models enhances glomerular damage while its deletion protects against disease progression [40]. Furthermore, studies in cultured primary renal tubular epithelial cells (TECs) showed administration of the saturated fatty acid palmitate resulted in an upregulation of osteopontin and CD44. These findings support the significance of CD44 and osteopontin expression in fatty acid-induced tubular cell damage in DKD [41]. We identified no studies directly connecting SHMT2 to DKD. Nonetheless, Shmt2 expression has been implicated in oxidative stress [42,43]. Oxidative stress is a known contributor to insulin resistance and the pathogenesis of DKD [44].

Our investigation has several limitations. Foremost amongst these limitations is the small human sample size. This sample size is not large enough to correct for clinical covariates. Therefore, orthogonal datasets in the mouse were used to increase the confidence in our results. In the human analysis, only a small number of genes passed an FDR-corrected level of significance. Accordingly, the analysis emphasized network-based analyses of pathways that were corrected for multiple testing corrections. In the murine analysis, rapidly acquired fresh tissue is required for both polyribosomal profiling and metabolomics. Thus, we did not examine translation or metabolism specific to the glomerulus or proximal tubule. Translation-related pathways were not dysregulated in the snRNAseq data, potentially due to the fact that snRNAseq expression is measured in the nucleus as opposed to the cytoplasm. Two pathway enrichment methodologies were used: a Fisher’s exact test and a transcriptogram network analysis, each of which identified different pathways. The Fisher’s exact test emphasizes the distribution of DEGs above a significance threshold. A transcriptogram network analysis balances magnitude, the direction of effect, significance, and protein–protein interactions of all protein-coding genes. Finally, an additional limitation was the use of male mice at multiple ages. We did not have the sample size in the human cohort to examine age or sex differences.

In conclusion, we utilized a multi-omics approach to demonstrate that alterations in protein translation and carboxylic acid catabolism are key features in both human DKD and a murine model of DKD, thus underscoring the value of murine models in the study of DKD. Interestingly, these alterations are prominent in the proximal tubule which highlights the importance of tubular dysfunction in the pathophysiology of DKD. In addition, we identified prominent genes expressed in the carboxylic acid catabolism pathway from the proximal tubule that were associated with more rapid eGFR loss in DKD. This finding provides promise for the development of novel markers of DKD progression as well as new therapeutic targets.

## Figures and Tables

**Figure 1 cells-11-01166-f001:**
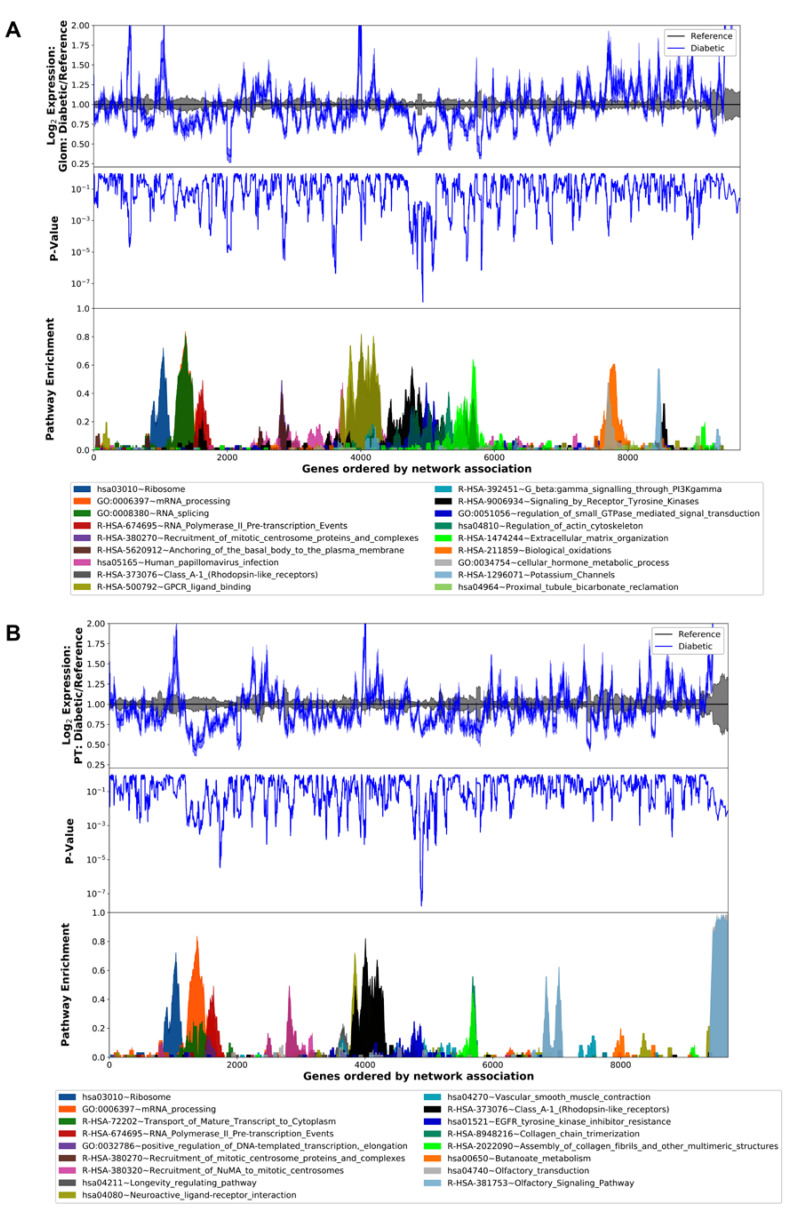
Transcriptogram of pathway enrichment between human diabetic and reference kidney sub-segments. X-axis: Genes ordered by network association. Y-axis: (top) Log_2_ fold change expression between diabetic and reference (**A**) glomerulus and (**B**) proximal tubules. (mid) gene expression *p*-values. (bottom) pathway enrichment score ranging from 0–1.

**Figure 2 cells-11-01166-f002:**
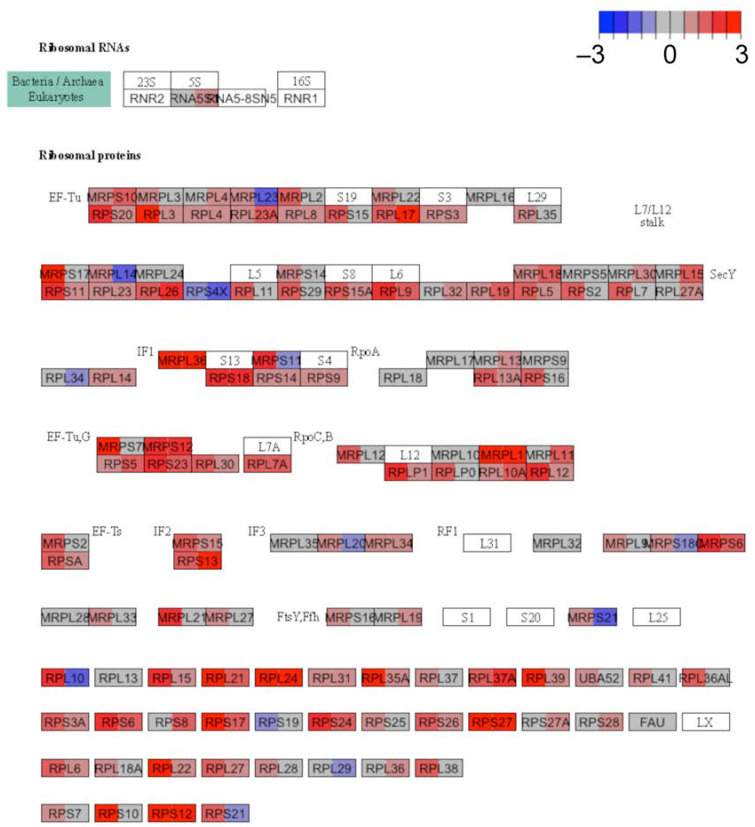
Ribosome pathway schematic highlighting gene expression changes in diabetic kidney disease. Gene expression changes between diabetic and reference kidneys are mapped to the KEGG ribosome pathway. Each box represents the fold change in expression between diabetic and reference in the glomerulus (left half of box) and proximal tubules (right half of box). Red indicates increased expression in diabetics. Blue indicates decreased expression in diabetics. White boxes indicate that genes involved in the subunit were not detected.

**Figure 3 cells-11-01166-f003:**
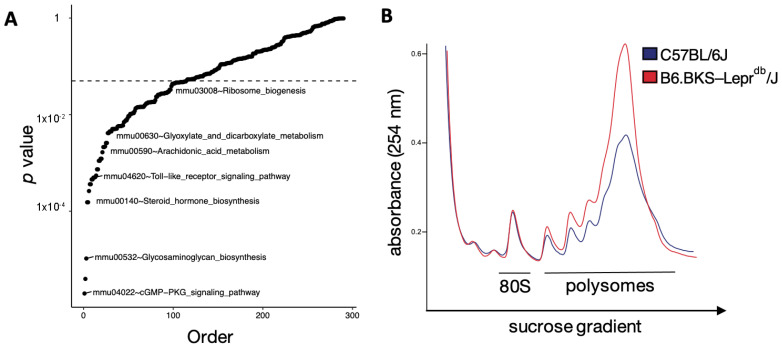
Translation is upregulated in the kidneys of diabetic mice. (**A**) Transcriptogram pathway analysis between diabetic and control mice. X-axis: Pathways ordered by *p*-values. Y-axis: −Log_10_ of *p*-values. (**B**) Polysomal profiling of kidney extracts from diabetic and control mice. X-axis: Increasing concentration of sucrose gradient. Y-axis: Absorbance of RNA at 254 nanometers.

**Figure 4 cells-11-01166-f004:**
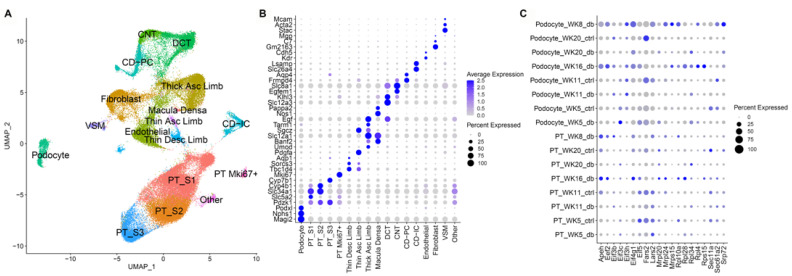
Single nuclear-sequencing analysis of kidney tissue between diabetic and control mice. (**A**) Uniform Manifold Approximation Projection (UMAP) of 17 kidney sub-segments. Dot plot highlighting selected genes used to (**B**) classify sub-segments and (**C**) representative genes altered in translation between diabetic and controls.

**Figure 5 cells-11-01166-f005:**
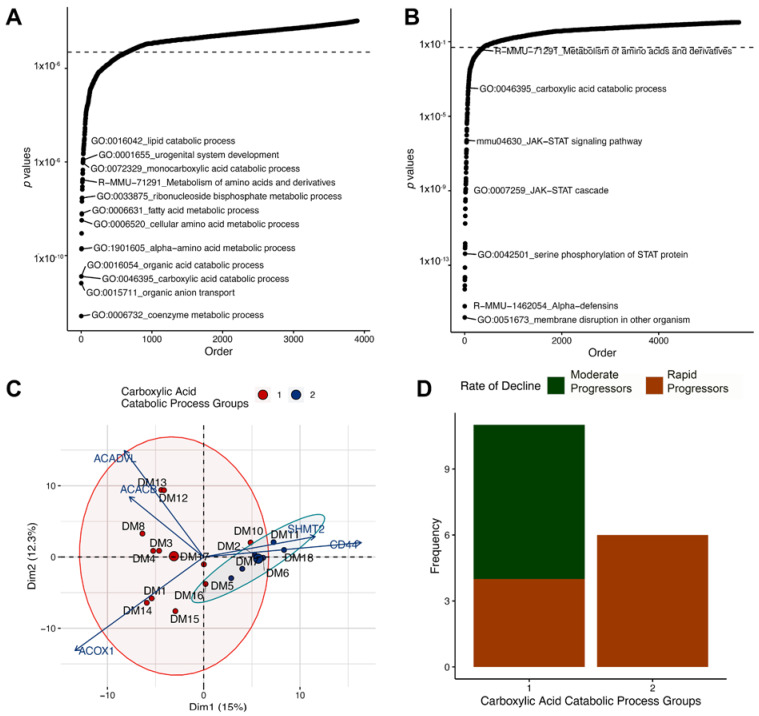
Carboxylic acid catabolic process gene expression associated with increased progression of diabetes. Pathway analysis between diabetic and controls in the proximal tubules of (**A**) mice and (**B**) humans. X-axis: Pathways ordered by *p*-values. Y-axis: −Log_10_ of *p*-values. (**C**) Principal component analysis of human diabetic proximal tubule samples clustered with carboxylic acid catabolic process genes using partitioning around medoids. Arrows represent the degree of impact and direction of relationship for the top five genes influencing the clusters. (**D**) Bar graph depicting the frequency of moderate progressors and rapid progressors, as defined by rate of decline in kidney function, within the two carboxylic acid groups identified in (**C**).

**Figure 6 cells-11-01166-f006:**
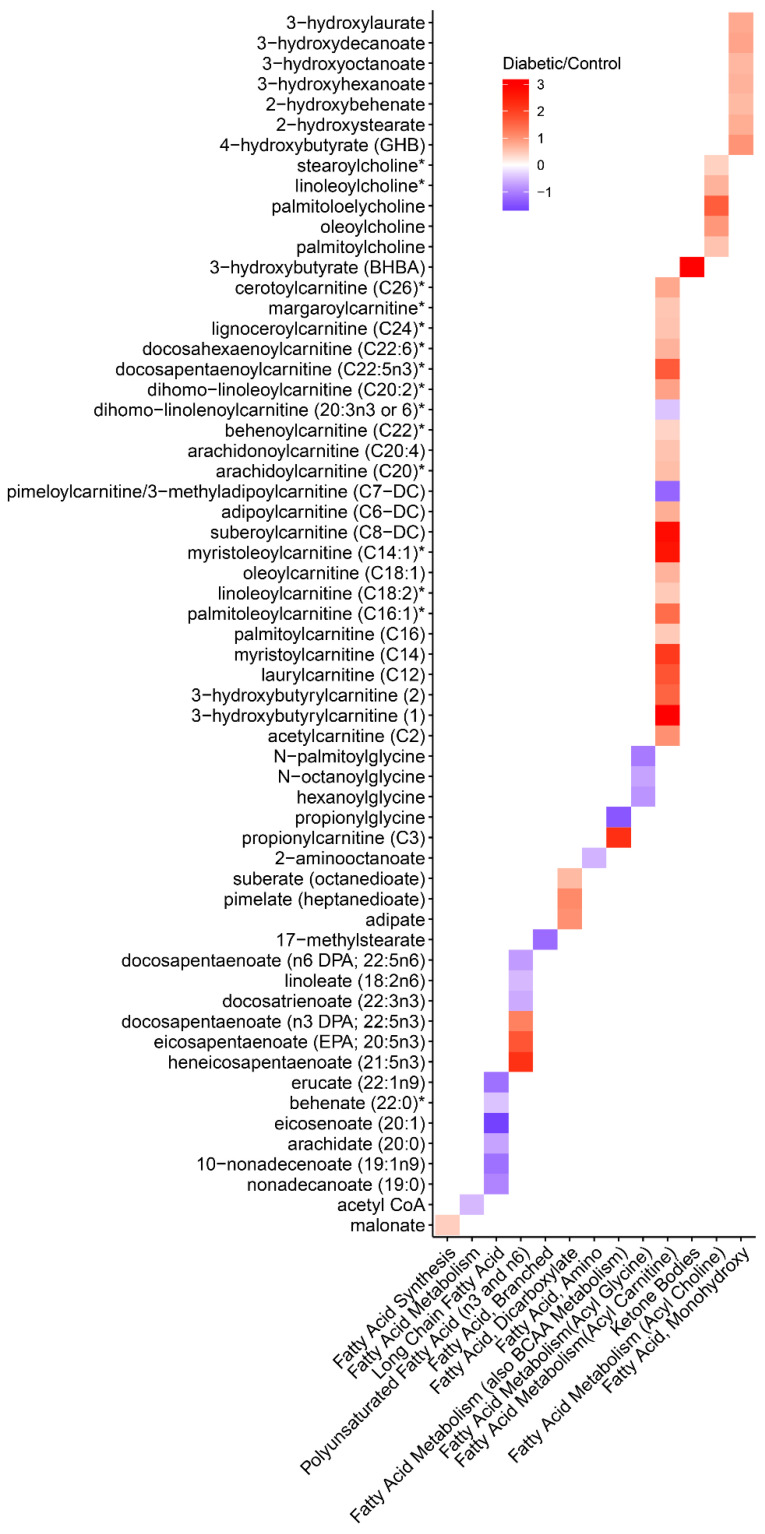
Heatmap of differentially regulated metabolites profiled between diabetic and control mice in fatty acid pathways. X-axis: Biochemicals. Y-axis: Fatty Acid Pathways. Legend represents the log_2_ fold changes with red meaning higher concentrations in diabetics and blue meaning lower concentrations in diabetics compared to controls. * Indicates annotated compounds without official confirmation based on a standard.

**Table 1 cells-11-01166-t001:** Summary clinical characteristics of samples.

Variable Mean ± SD or *n* (%)	Diabetic Kidney Disease *n* = 18	Reference *n* = 9
Age	55.0 ± 9.3	50.0 ± 14.2
Gender, Female	11 (61.1)	6 (66.7)
Race, Black ^A^	5 (27.8)	NA
Baseline eGFR (mL/min) ^B^	63.2 ± 26.2	NA
Baseline proteinuria >3 gm	10 (55.6)	NA
Endpoint eGFR (mL/min)	22.4 ± 19.2	NA
Rate of progression (mL/min/year)	14.5 ± 19.6	NA
Patients with > 10 mL/min/yr GFR loss	11 (61.1)	NA
Duration of follow-up data (months)	47.7 ± 23.7	NA
Histopathologic diabetic kidney disease	18 (100)	0 (0)
Histopathologic arterionephrosclerosis	18 (100)	0 (0)
Glomerular obsolescence (% of glomeruli affected)	24.7 ± 18.7	23.9 ± 21.2
IFTA (% of cortex affected)	48.9 ± 12.8	20.0 ± 13.8
Arteriolar Hyalinosis severity (scale of 0–3)	2.4 ± 0.6	0.9 ± 0.4
Presence of nodular glomerulosclerosis	15 (83.3)	0 (0)
Presence of effacement (>80% of foot processes)	9 (50)	0 (0)

^A^ Other subjects were white (*n* = 12) and other, not specified (*n* = 1). ^B^ The baseline eGFR was defined as the pre-biopsy eGFR value within a year of biopsy, but prior to any decline of 20% or greater in eGFR. IFTA—interstitial fibrosis and tubular atrophy. eGFR—estimated glomerular filtration rate, according to the CKD-EPI equation, NA—not available.

## Data Availability

Transcriptomic animal data is located at: https://connect.rstudio.iu.edu/content/21/ accessed 23 January 2022. Human transcriptomic data is available in the Kidney Precision Medicine Project (KPMP) Atlas https://atlas.kpmp.org/repository/ accessed 23 January 2022.

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
