# Peer review of "Alterations in Protein Translation and Carboxylic Acid Catabolic Processes in Diabetic Kidney Disease"

_cells, 2022, doi:10.3390/cells11071166_

Round 1
Reviewer 1 Report
The authors utilized a multi-omics approach and demonstrated that alterations in protein translation and carboxylic acid catabolism are key features in both human DKD and in a murine model of DKD. These alterations seem to be more prominent in the proximal tubule which shows the importance of tubular dysfunction in DKD.
The results of the study are important and novel; I have the following comments:
- what was the reason for kidney biopsy? where there any other lesions that suggest another diagnosis? frequntly patients with DKD who undergo kidney biopsy have another superimposed glomerular disease.
- the presumed role of carboxylic acid catabolism pathway in DKD pathogeneis should be better described in the discussion section.
- A figure with the study design and workflow should be presented in the manuscript.
- the work of Zhou et al doi: 10.1093/ckj/sfaa190 - should be discussed
Author Response
Please see the attached word document.

Reviewer 2 Report
Current Manuscript, authors took advantage of multi-omics approach to demonstrate the alterations in protein translation (glomeruli and proximal tubules) and carboxylic acid catabolism in both human DKD and a murine model of DKD. LMD transcriptomics and Single nuclei RNAseq from human and mouse kidney data sets overlap analysis was performed. Authors found that, several pathways altered includes kidney developmental pathway in glomeruli and carboxylic acid catabolic process, steroid hormone biosynthesis, glutathione metabolism, and biological oxidations were enriched within the proximal tubule cells. Basis of snRNAseq in PT cells and LMD from Human Kidney PT compartment, the major pathway identified in the mouse PT cells that was also enriched in the human dataset was carboxylic acid catabolic process. Metabolomics from mouse kidneys indicated the dysregulation of metabolites related to fatty acid metabolism and ketone pathway. Authors indicated that Acetyl-CoA was decreased in DB kidney though the downregulation seems to be very minimal. Current study explored the translational, transcriptomic signatures from both human and mouse model of DKD. The experimental approach and study design are appropriate and well performed. However, authors used the mouse model (#000697 db mice from Jackson Labs) is not well established or reliable/suitable and mimic the condition (phenotypic changes) of diabetic kidney disease is very minimal. I wonder why authors choose DB mice from C57BL6 background instead #000642 is DB mice, a well-established mouse model of DKD using by many investigators (Matthew D. Breyer et al 2005, JASN) actively working on Diabetic kidney disease (DKD). Therefore, several major concerns need to be address by authors.
Major comments:
- One of the major concerns is mouse model (#000697) that has been used in this current study. Based on evidences, the albuminuria, blood glucose, TG and Cholesterol along with pathophysiology changes were very minimal or subtle in this DB strain and may not reliable to compare the phenotypic changes with human patients with DKD. However, authors did not present any data related to biochemistry and pathology from this mouse strain.
- In the current study, authors used Male DB mice to compare the data from the Female Diabetic Patients. Majority of Human Kidney sections were from Female patients (61.1%). Authors need to address the gender differences.
- In methods, authors indicated that mice were sacrificed and kidneys harvested (for snRNAseq experiment) at the age of 5-20 weeks. The age difference is very long which will have impact on the pathogenesis of DKD at early and late on set of disease progression and results will be affect greatly. For any such studies, the age of the experimental mice very important and differentially effect the phenotypic changes and disease progression when compare with their age matched control.
- Based on the LMD analysis from human kidney sections and snRNAseq of DB mouse kidneys, authors found any difference in transcriptome signatures related to inflammation? As inflammation of one of the important factors involved in the progression of DKD.
- In methods Line 142, authors indicated that Bulk RNA sequencing was performed. However, I did not see any data sets related to Bulk RNA seq? I believe, I didn’t miss it.
- In the Result Part, Section 3.4 and 3.5 (Line 335-379 and Line 386-421) looks exactly similar to each other. Needs to rewrite completely.
Author Response
Please see the attached word document

Round 2
Reviewer 2 Report
The current Manuscript is revised version and is important in the field of Diabetic Kidney Disease (DKD) as authors focused on the differential expression in protein translations and Carboxylic Acid Metabolism in both rodent model and patients with DKD. The results showed from SnRNAseq and transcriptogram network analysis. The authors addressed all the comments reasonably in response to all the comments and questions raised. All the responses and changes were made in this current revised Manuscript is appropriate and can be accepted as Manuscript in the Journal of Cells, MDPI. Congratulations to all the authors.